# Validating a popular outpatient antibiotic database to reliably identify high prescribing physicians for patients 65 years of age and older

Kevin L. Schwartz[1,2,3,4]*, Cynthia Chen[2], Bradley J. Langford[1], Kevin A. Brown[1,2,3], Nick Daneman[1,2,5,6], Jennie Johnstone[1,3,7], Julie HC Wu[1], Valerie Leung[1], Gary Garber[1,8]

**1** Public Health Ontario, Toronto, Ontario, Canada, **2** ICES, Toronto, Ontario, Canada, **3** Dalla Lana School of Public Health, University of Toronto, Toronto, Ontario, Canada, **4** Unity Health Toronto, Toronto, Ontario, Canada, **5** Sunnybrook Research Institute, Division of Infectious Diseases, Toronto, Ontario, Canada, **6** The Institute of Health Policy, Management, and Evaluation, University of Toronto, Ontario, Canada, **7** Sinai Health System, Toronto, Ontario, Canada, **8** Ottawa Research Institute, Ottawa, Ontario, Canada

* Kevin.schwartz@oahpp.ca

**Data Availability Statement:** The data for this study is not publically available. The data set from this study is held securely in coded form at ICES. While data sharing agreements prohibit ICES from

## Abstract

### Objective

Many jurisdictions lack comprehensive population-based antibiotic use data and rely on third party companies, most commonly IQVIA. Our objective was to validate the accuracy of the IQVIA Xponent antibiotic database in identifying high prescribing physicians compared to the reference standard of a highly accurate population-wide database of outpatient anti-microbial dispensing for patients ≥65 years.

### Methods

We conducted this study between 1 March 2016 and 28 February 2017 in Ontario, Canada. We evaluated the agreement and correlation between the databases using kappa statistics and Bland-Altman plots. We also assessed performance characteristics for Xponent to accurately identify high prescribing physicians with sensitivity, specificity, positive predictive value (PPV), and negative predictive value.

### Results

We included 9,272 physicians. The Xponent database has a specificity of 92.4% (95%CI 92.0%-92.8%) and PPV of 77.2% (95%CI 76.0%-78.4%) for correctly identifying the top 25th percentile of physicians by antibiotic volume. In the sensitivity analysis, 94% of the top 25th percentile physicians in Xponent were within the top 40th percentile in the reference database. The mean number of antibiotic prescriptions per physician were similar with a relative difference of -0.4% and 2.7% for female and male patients, respectively. The error was greater in rural areas with a relative difference of -8.4% and -5.6% per physician for female and male patients, respectively. The weighted kappa for quartile agreement was 0.68 (95% CI 0.67–0.69).

making the data set publicly available, access may be granted to those who meet pre-specified criteria for confidential access, available at www.ices.on. ca/DAS. The full data set creation plan and underlying analytic code are available from the authors upon request, understanding that the programs may rely upon coding templates or macros that are unique to ICES. The IQVIA Xponent dataset is owned and proprietary by IQVIA. The contract between PHO and IQVIA does not permit us to share the data publically. The authors had no special access privileges and other researchers may purchase the data from IQVIA directly (www. iqvia.com).

**Funding:** This research was supported by Public Health Ontario. This study was also supported by ICES, which is funded by an annual grant from the Ontario Ministry of Health and Long-Term Care (MOHLTC). The opinions, results and conclusions reported in this paper are those of the authors and are independent from the funding sources. No endorsement by ICES or the Ontario MOHLTC is intended or should be inferred. The funders had no role in study design, data collection and analysis, decision to publish, or preparation of the manuscript.

**Competing interests:** The authors have declared that no competing interests exist.

## Conclusion

We validated the IQVIA Xponent antibiotic database to identify high prescribing physicians for patients ≥65 years, and identified some important limitations. Collecting accurate population-based antibiotic use data will remain vital to global antimicrobial stewardship efforts.

## Introduction

Rising antimicrobial resistance is a global public health threat jeopardizing multiple advances in modern medicine.[1] Antibiotic overuse in humans is associated with resistance at both the individual [2] and ecological [3] levels, and is the most important modifiable driver of antimicrobial resistance. An essential component of global, national, and regional antimicrobial stewardship strategies is improved surveillance of antibiotic use.[4–6]

Various forms of antibiotic use data are available including dispensing data compiled by pharmacies (e.g. National Prescription Audit® CompuScript® or GPM®)[7] [8] or funders (e.g. Ontario Drug Benefit) [9] which can be population based. Other databases include dispensed data at the prescriber level (e.g. Xponent®) [10] or by prescription from electronic medical records.[11] These data sources have important differences and may not correlate over time. [11,12] Many jurisdictions do not have access to valid population-based community antibiotic use data and rely on third party proprietary data. The most common source of antibiotic use data globally is IQVIA, formerly IMS Health. IQVIA maintains a variety of prescription drug databases. In the United States (U.S.) and Canada, the IQVIA Xponent database is derived from 65–70% of the populations' dispensed prescriptions [7,13] IQVIA then applies a proprietary geospatial extrapolation algorithm. The data are then sold to third parties representing 100% complete prescription data. The projection methodology is internally validated by the company.[14] Despite some uncertainty in the data validity multiple organizations in Europe, Asia, and North America, such as the U.S. Centers for Disease Control and Prevention and Health Canada, rely on these data for both research and surveillance of antibiotic use patterns.[7,13–23] In Canada we have used IQVIA data to describe overall antibiotic use,[7,24] to identify predictors of prolonged antibiotic durations,[10] and we are currently studying the impact of providing audit and feedback letters from these data to primary care physicians (https://www.clinicaltrials.gov/ct2/show/NCT03776383).

The Ontario Drug Benefit (ODB) database has population-based drug dispensing data for medications funded by the ODB program, for all persons in Ontario, Canada over 65 years of age and has been previously validated compared to chart abstraction to be over 99% accurate. [25] Our objective in this study was to validate the accuracy of the IQVIA antibiotic Xponent database to identify high prescribing physicians compared to ODB as the gold standard for patients 65 years of age and older.

## Methods

### Setting

We conducted this study at ICES (formerly the Institute for Clinical Evaluative Sciences) in Ontario, Canada. ICES is a not-for-profit research institute and a prescribed entity under Ontario's *Personal Health Information Protection Act* with permission to securely collect and store personal health information. Ontario has universal health insurance which includes virtually all residents (excluding recent migrants within the previous 3 months, those residing on

an indigenous reserve, and military personnel). All persons 65 years of age or older, as well as select low income individuals or those on disability, have publically funded drug insurance through the ODB which includes most commonly used antibiotics.[26] The study population consisted of physicians prescribing antibiotics to patients 65 years of age or older, between 1 March 2016 and 28 February 2017. We have previously observed large disparities in antibiotic prescribing to male and female patients 65 years of age and older,[7] therefore we sought to validate physician prescribing to male and female patients separately.

### Data sources

Xponent is an IQVIA database with dispensed antibiotic prescription counts aggregated at the physician prescriber-level. Non-physician prescribers (i.e. dentists, nurse practitioners, optometrists, etc.) are not included in this database. Antibiotic prescription counts were provided for total and 13 antibiotic class specific groupings. Antibiotics were defined as World Health Organization Anatomical Therapeutic Chemical Classification J01 antibacterials for systemic use limited to those taken orally and dispensed from an outpatient pharmacy (S1 Table). Counts of antibiotic prescriptions were limited to outpatient new prescriptions, excluding refills, topical, and intravenous medications. The data also includes antibiotic prescribing rates (number of antibiotic prescriptions/number of total prescriptions by that physician) and proportion of antibiotic prescriptions that were prolonged in duration which we defined as >8 days.[10] The data was broken down by patient age and sex strata; all patients, males <18 years, females <18 years, males 18–64 years, females 18–64 years, males >64 years, and females >64 years. We limited this study to males and females >64 years for which we had complete population data in ODB to act as the reference standard.

IQVIA creates the Xponent database by obtaining prescription data directly from 2,187 (49.8%) of Ontario's 4391 pharmacies. As of 2017 this amounted to 61.3% of all Ontario prescriptions. IQVIA then incorporates sales and insurance data, as well as the geographical location of pharmacies not captured, into a patented geospatial projection algorithm and extrapolates to estimate all physician prescribed antibiotics. The methodology is proprietary, but according to the company is routinely internally validated.[15,27] Xponent was linked to ICES using unique College of Physicians and Surgeons of Ontario numbers.

We used ODB, held at ICES, from persons ≥65 years of age as the reference standard. The ODB database contains all dispensed medication claims prescribed by physicians from a formulary of 4400 prescription drugs funded through the ODB program. The ODB database has been previously validated against pharmacy chart abstraction and determined to be >99% accurate.[25] To exclude refilled prescriptions from both Xponent and ODB, we defined refills in ODB as the same drug being dispensed for the same patient, prescribed by the same physician for the identical dose and duration within 14 days of the expected end date of the previous prescription. The only antibiotic expected to be potentially poorly captured in ODB was doxycycline as was not funded by the ODB at the time of this study. We used the Ontario Health Insurance Plan database to ascertain the number of outpatient visits.

We excluded physicians who prescribed <9 antibiotics to patients ≥65 years of age during the study year in either Xponent or ODB databases. We used this cut off, which removes 5% of all antibiotic prescriptions, in order to exclude infrequent antibiotic prescribers. We excluded physicians with <200 total outpatient visits in the year based on billing claims in the Ontario Health Insurance Plan database. We also excluded the few physicians who had illogical antibiotic prescribing rates of ≤0% or ≥100%, or if there were more antibiotic prescriptions than patient visits.

## Statistical analysis

We performed multiple analyses to evaluate the agreement and correlation between Xponent and ODB antibiotic data in patients ≥65 years of age. Our primary interest with the data was to be able to reliably identify high antibiotic prescribing physicians. We organized both Xponent and ODB in quartiles based on the volume of antibiotic prescriptions per physician. We first dichotomized the quartiles as high (top quartile) and low (bottom three quartiles) prescribers to calculate the performance characteristics of accurately identifying a high prescriber in Xponent. We first did this for all physicians and then restricted to primary care physicians only since this specialty prescribes majority of the outpatient antibiotics.[24] We reported sensitivity, specificity, positive predictive value (PPV), and negative predictive value (NPV), with 95% confidence intervals (CIs). As a sensitivity analysis, we adjusted the definition of high prescribers in ODB from the top 25% to 30% and to 40% in order to evaluate the degree of Xponent error. For instance, if 20% of high prescribers were misclassified in Xponent, but 90% of these were in the top 40% of ODB, this provides more reassurance related to the magnitude of error in the data. If low prescribing physicians in the bottom 60% were misclassified as high prescribers, this would represent a more clinically significant misclassification.

We descriptively compared the mean number of antibiotics prescribed per physician. Then we used the quartiles and evaluated agreement using weighted kappa statistics with 95% CIs for total antibiotics as well as subclasses of antibiotics. We did this for male and female patients separately as well as by urban and rural locations of practice. Rural areas were defined by the physician location of practice residing in a town with a population of <10,000. Kappa statistics were considered poor if <0, slight if 0–0.20, fair if 0.21–0.40, moderate if 0.41–0.60, substantial if 0.61–0.80, and almost perfect if 0.81–1.00.[28] To quantify the magnitude of the error we calculated the relative difference in the number of antibiotics per physician as (Xp-ODB)/ODB x100; where Xp is the number of antibiotics per physician in Xponent and ODB is the number of antibiotics per physician in the ODB database.

We evaluated the total antibiotic prescribing rate (number of antibiotic prescriptions per 100 total prescriptions) using Bland-Altman plots.[29] Bland-Altman plots allow one to visualize the error between two data sources by plotting the difference of the values on the Y-axis (Xponent-ODB) against the mean of the values on the X-axis ((Xponent+ODB)/2), along with plots of the two standard deviation (SD) limits above and below the mean difference. We performed this separately for male and female patients. IQVIA data does not contain numbers of patient visits, therefore we assessed the correlation of the antibiotic prescribing rate (number of antibiotic prescriptions per 100 total prescriptions) to number of antibiotic prescriptions per 100 patient visits using Spearman correlation coefficients.

To validate the antibiotic duration data, we defined antibiotic prescription durations that were >8 days as prolonged.[10] Similar to the above analysis we organized physicians into quartiles of proportion of prolonged durations and calculated performance characteristics, weighted kappa statistics, and Bland-Altman plots as described above for all physicians and primary care physicians only. Results were reported in accordance with guidelines for validation studies with administrative data.[30]

## Ethics

This project has Ethics Research Board approval from Public Health Ontario 2017–032.02.

## Results

The Xponent database had 25,678 unique physicians and 24,878 (97%) were successfully linked to ODB at ICES using College of Physicians and Surgeons identifiers. After applying

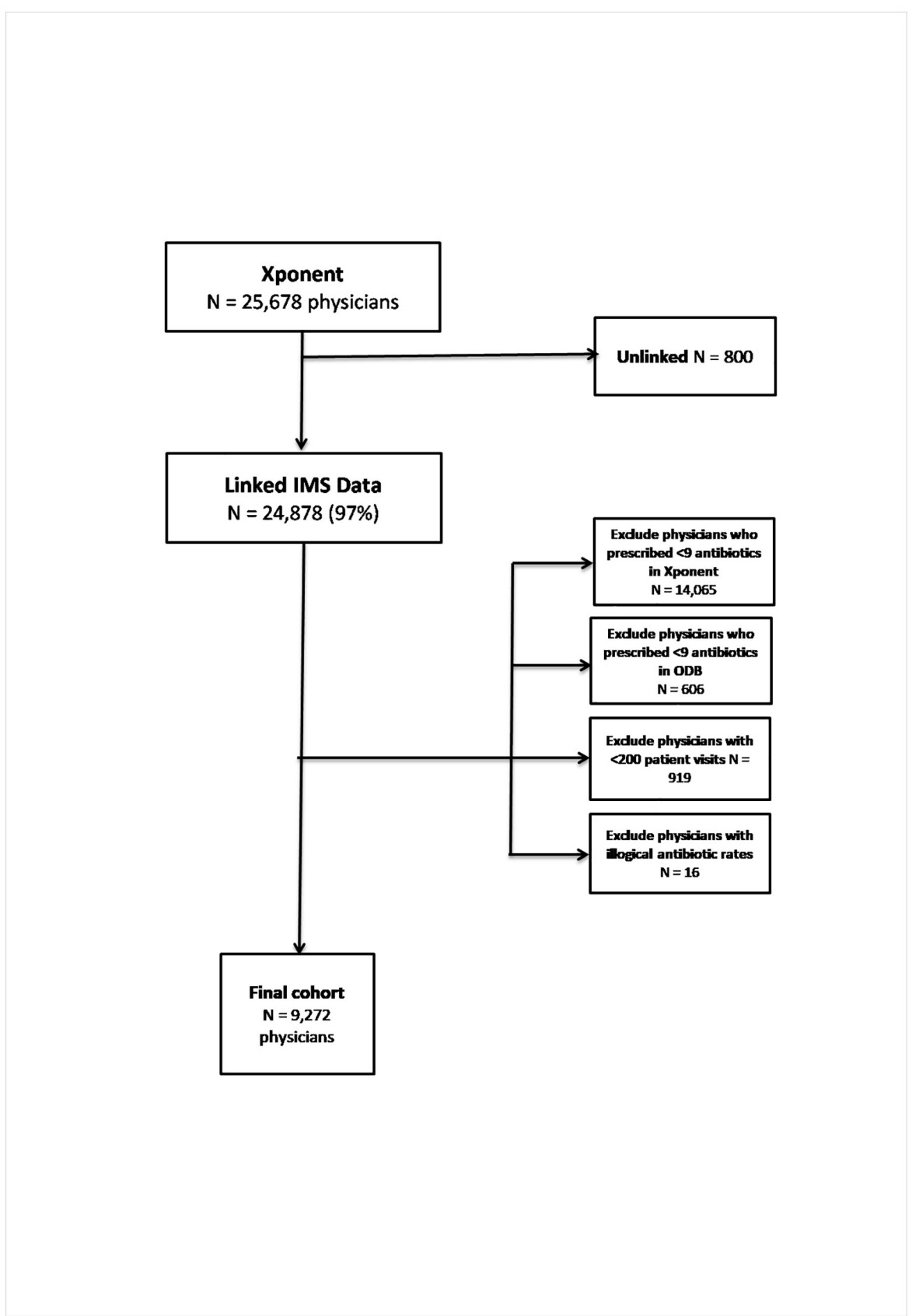

**Fig 1. Flow chart of study exclusions.**

**Table 1. Demographic characteristics of the 9,272 physicians included in this study.**

| Physician characteristic | Number (%) |
|---|---|
| Age, mean (SD)* | 52.3 (11.4) |
| Years since medical graduation^ | |
| Early career (<11 years) | 3,309 (35.7%) |
| Middle career 11–24 years) | 2,396 (25.8%) |
| Late career (≥25 years) | 3,567 (38.5%) |
| Gender^ | |
| Female | 3,000 (32.4%) |
| Male | 6,272 (67.6%) |
| Specialty^ | |
| Primary care** | 7,734 (83.3%) |
| Internal medicine and subspecialties | 669 (7.3%) |
| Surgical specialty | 562 (6.1%) |
| Emergency medicine | 200 (2.2%) |
| Dermatology | 68 (0.7%) |
| Other | 39 (<0.1%) |
| Area of practice | |
| Urban | 8,376 (90.3%) |
| Rural | 896 (9.7%) |
| Location of medical training | |
| Canada | 6,093 (65.7%) |
| Outside Canada | 2,346 (25.3%) |
| Missing | 833 (9.0%) |
| Median daily patient visits | |
| 0–9 | 34 (0.4%) |
| 10–19 | 2,035 (22.0%) |
| ≥20 | 7,203 (77.7%) |
| Average patient Chronic Disease Score per physician*** | |
| Low (≤4) | 994 (10.7%) |
| Medium (5) | 3,504 (37.8%) |
| High (≥6) | 4,770 (51.5%) |

*8.8% of physicians were missing a value for age

**Primary care defined as specialties of family medicine, general practice, or community medicine

***The Chronic Disease Score is a comorbidity index based on pharmaceutical data,[34] Missing data from 4 physicians. ^These variables derived from Xponent, the other variables are from ICES.

the study exclusion criteria 14,671 were excluded for prescribing <9 antibiotics, 919 for <200 patient visits, and 16 for illogical prescribing rates, leaving 9,272 physicians included in this validation analysis (Fig 1).

## Physician characteristics

These physicians were mostly male (68%) and 83% were primary care physicians. The majority of these physicians saw ≥20 patients daily, 90% worked in an urban area, and 34% were not Canadian trained (Table 1).

## Performance characteristics

The Xponent database has a specificity of 92.4% (95%CI 92.0%-92.8%) and PPV of 77.2% (95%CI 76.0%-78.4%) for correctly identifying the top 25th percentile of physicians by antibiotic prescribing volume. The performance measures of correctly identifying the top 25th

**Table 2. Performance characteristics of Xponent in defining the top quartile (25% of physicians) as compared to the reference standard of the Ontario Drug Benefit (ODB) for antibiotic volume and the proportion of prolonged duration prescriptions.**

| Comparator in ODB | TP | FN | FP | TN | Sensitivity (95% CI) | Specificity (95% CI) | Positive Predictive Value (95% CI) | Negative Predictive value (95% CI) |
|---|---|---|---|---|---|---|---|---|
| **Comparing the highest 25% in IQVIA to the highest 25% in ODB** | | | | | | | | |
| All physicians antibiotic volume | 3580 | 1056 | 1056 | 12852 | 77.2% (76.0%-78.4%) | 92.4% (92.0%-92.8%) | 77.2% (76.0%-78.4%) | 92.4% (92.0%-92.8%) |
| Family physicians antibiotic volume | 2924 | 942 | 942 | 10660 | 75.6% (74.3%-77.0%) | 91.9% (91.4%-92.4%) | 75.6% (74.3%-77.0%) | 91.9% (91.4%-92.4%) |
| All physician proportion prolonged duration | 3799 | 837 | 837 | 13071 | 82.0% (80.8%-83.0%) | 94.0% (93.6%-94.4%) | 82.0% (80.8%-83.0%) | 94.0% (93.6%-94.4%) |
| Family physician proportion prolonged duration | 3173 | 693 | 693 | 10909 | 82.1% (80.8%-83.3%) | 94.0% (93.6%-94.5%) | 82.1% (80.8%-83.3%) | 94.0% (93.6%-94.5%) |
| **Sensitivity analysis** | | | | | | | | |
| **Comparing the highest 25% in IQVIA to the highest 30% in ODB** | | | | | | | | |
| All physicians antibiotic volume | 3949 | 1613 | 687 | 12295 | 71% (69.8%-72.2%) | 94.7% (94.3%-95.1%) | 85.2% (84.1%-86.2%) | 88.4% (87.9%-88.9%) |
| Family physicians antibiotic volume | 3240 | 1400 | 626 | 10202 | 69.8% (68.5%-71.2%) | 94.2% (93.8%-94.7%) | 83.8% (82.6%-85.0%) | 87.9% (87.3%-88.5%) |
| All physician proportion prolonged duration | 4126 | 1436 | 510 | 12472 | 74.2% (73.0%-75.3%) | 96.1% (95.7%-96.4%) | 89.0% (88.1%-89.9%) | 89.7% (89.2%-90.2%) |
| Family physician proportion prolonged duration | 3431 | 1209 | 435 | 10393 | 73.9% (72.7%-75.2%) | 96.0% (95.6%-96.3%) | 88.8% (87.7%-89.7%) | 89.6% (89.0%-90.1%) |
| **Comparing the highest 25% in IQVIA to the highest 40% in ODB** | | | | | | | | |
| All physicians antibiotic volume | 4371 | 3045 | 265 | 10863 | 58.9% (57.8%-60.1%) | 97.6% (97.3%-97.9%) | 94.3% (93.6%-94.9%) | 78.1% (77.4%-78.8%) |
| Family physicians antibiotic volume | 3601 | 2585 | 265 | 9017 | 58.2% (57.0%-59.5%) | 97.2% (96.8%-97.5%) | 93.2% (92.3%-93.9%) | 77.7% (77.0%-78.5%) |
| All physician proportion prolonged duration | 4421 | 2995 | 215 | 10913 | 59.6% (58.5%-60.7%) | 98.1% (97.8%-98.3%) | 95.4% (94.7%-96.0%) | 78.5% (77.8%-79.2%) |
| Family physician proportion prolonged duration | 3666 | 2520 | 200 | 9082 | 59.3% (58.0%-60.5%) | 97.9% (97.5%-98.1%) | 94.8% (94.1%-95.5%) | 78.3% (77.5%-79.0%) |

TP = True Positives; FN = False Negatives; FP = False Positives; TN = True Negatives; Prolonged duration defined as >8 days

percentile of prolonged duration prescribers was slightly better with a specificity of 94.0% (95%CI 93.6%-94.4%) and PPV of 82.0% (95%CI 80.8%-83.0%). There was little difference when the cohort was limited to only primary care physicians (Table 2). The sensitivity analysis demonstrated that the magnitude of the misclassification was relatively small. In the primary analysis 77% of physicians were correctly classified in Xponent as top 25th percentile prescribers, however 85% were at least within the top 30th percentile, and 94% in the top 40th percentile of ODB prescribers (Fig 2).

## Agreement

We descriptively compared the mean number of antibiotic prescriptions per physician and found them to be similar with a relative difference of -0.4% and 2.7% for female and male patients, respectively. The error was greater in rural areas with a relative difference of -8.4% and -5.6% per physician for female and male patients, respectively. There were some notable differences between antibiotic subclasses, particularly for trimethoprim and/or sulfamethoxazole

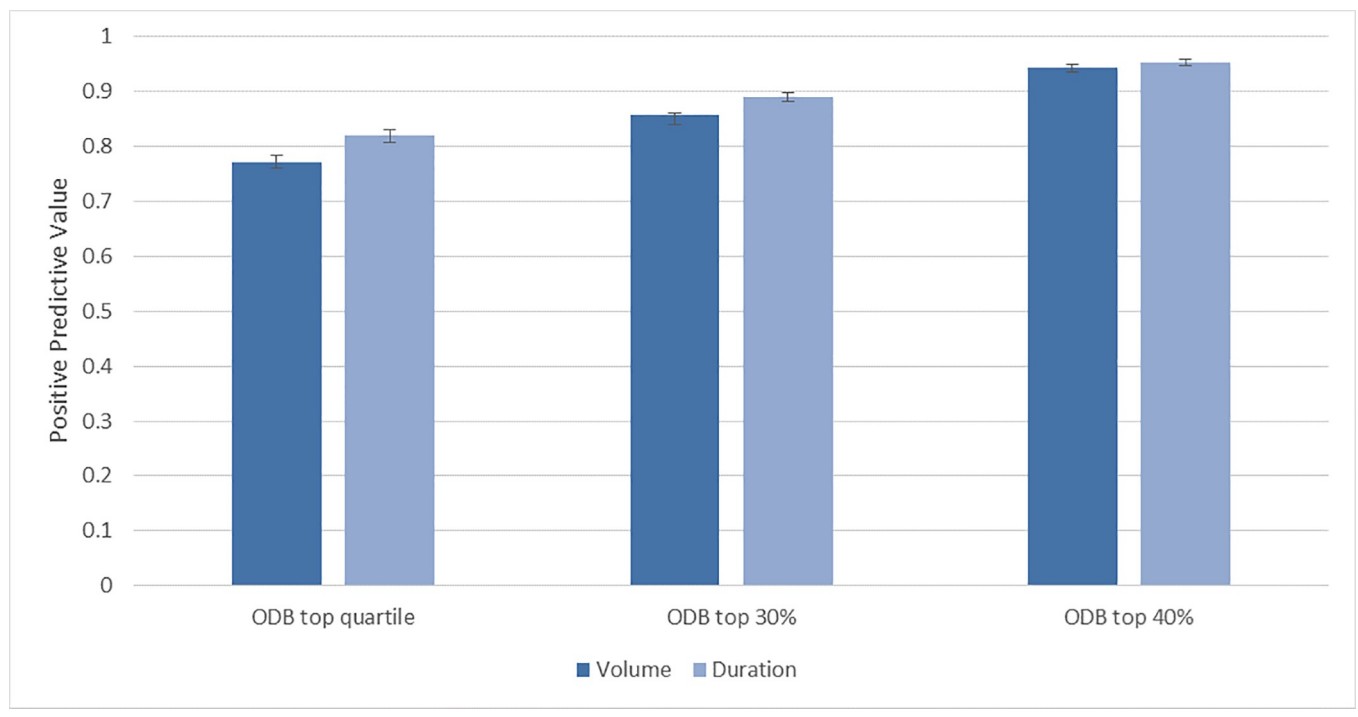

**Fig 2. Positive Predictive Values, with 95% confidence intervals, comparing the highest quartile from Xponent to ODB to identify high volume and high prolonged duration prescribers.**

(Table 3). Overall, the weighted kappa for quartile agreement between Xponent and ODB was substantial at 0.68 (95%CI 0.67–0.69) for both female and male patients.

We constructed Bland-Altman plots to visualize and quantify the agreement in antibiotic prescribing rate (number of antibiotic prescriptions per 100 total prescriptions) between Xponent and ODB (Fig 3). These databases had high agreement for this variable with a mean difference (Xponent rate minus ODB rate) of -0.5 antibiotics per 100 total prescriptions for female patients. Meaning, that on average the antibiotic rate was 0.5 antibiotics per 100 total prescriptions higher in ODB than Xponent. However there were substantial outliers (mean-2SD = -16.4 to mean+2SD 15.4). Results were similar for male patients (S1 Fig).

The proportion of prolonged duration antibiotic prescriptions had overall substantial agreement as well between the data sources with more error observed in rural areas (Table 4). On the Bland-Altman plot the mean difference (Xponent-ODB) was -0.3 prolonged antibiotic prescriptions per 100 antibiotic prescriptions. Meaning, that on average the proportion of prolonged antibiotic duration prescriptions was 0.3% higher in ODB than Xponent. Similarly, substantial outliers were observed (mean-2SD -17.8 to mean+2SD 17.1) (Fig 4 and S2 Fig).

Physician ranking according to antibiotic prescriptions per 100 total prescriptions in Xponent was strongly correlated to the number of antibiotic prescriptions per 100 patient visits in ODB data for primary care physicians treating female (Spearman coefficient = 0.93 p<0.001, Fig 5) and male (Spearman coefficient = 0.93, p<0.001, S3 Fig) patients.

## Discussion

Accurately monitoring population antibiotic use is a critical component of antimicrobial stewardship. We have validated that the IQVIA Xponent antibiotic database reliably identified high physician prescribers for patients 65 years of age and older. We identified that 23% of

**Table 3. Agreement between physician antibiotic prescribing quartiles in Xponent compared to the Ontario Drug Benefit (ODB) database.**

| Stratum | Mean number of antibiotic prescriptions per physician | | | Agreement between quartiles |
|---|---|---|---|---|
| | Xponent | ODB | Relative difference* | Weighted kappa (95% CI) |
| Female Patients | | | | |
| Total antibiotics | 85.2 | 85.5 | -0.4% | 0.68 (0.67–0.69) |
| Rural regions | 79.6 | 86.8 | -8.4% | 0.69 (0.65–0.72) |
| Urban regions | 85.8 | 85.4 | 0.5% | 0.68 (0.67–0.69) |
| Antibiotic subclasses | | | | |
| Penicillins without Beta-Lactamase Inhibitor | 11.8 | 11.8 | -0.5% | 0.71 (0.70–0.72) |
| Penicillins with Beta-Lactamase Inhibitor | 4.8 | 4.9 | -2.0% | 0.64 (0.63–0.65) |
| Cephalosporins (First Generation) | 8.9 | 9.3 | -4.0% | 0.68 (0.67–0.69) |
| Cephalosporins (Second/Third Generation) | 4.7 | 4.8 | -1.6% | 0.69 (0.68–0.70) |
| Fluoroquinolones (Second Generation) | 11.4 | 11.1 | 3.4% | 0.69 (0.68–0.70) |
| Fluoroquinolones (Third Generation) | 5.9 | 5.9 | 0.3% | 0.65 (0.64–0.66) |
| Macrolides | 12.8 | 12.7 | 1.1% | 0.73 (0.72–0.74) |
| Trimethoprim and/or Sulphonamides | 4.9 | 5.6 | -13.1% | 0.64 (0.62–0.65) |
| Nitrofurantoin | 12.8 | 13.8 | -7.4% | 0.71 (0.70–0.72) |
| Tetracyclines** | 1.5 | 0.6 | 167.5% | 0.42 (0.40–0.43) |
| Lincosamides | 1.4 | 1.4 | 0.3% | 0.70 (0.69–0.71) |
| Metronidazole | 2.1 | 2.1 | 1.5% | 0.67(0.66–0.68) |
| Other | 1.5 | 1.7 | -12.1% | 0.81 (0.80–0.82) |
| Male Patients | | | | |
| Total antibiotics | 57.9 | 56.3 | 2.7% | 0.68 (0.67–0.69) |
| Rural regions | 51.8 | 54.9 | -5.6% | 0.69(0.66–0.73) |
| Urban regions | 58.5 | 56.5 | 3.6% | 0.68 (0.66–0.69) |
| Antibiotic subclasses | | | | |
| Penicillins without Beta-Lactamase Inhibitor | 8.6 | 8.6 | 0.7% | 0.69 (0.68–0.70) |
| Penicillins with Beta-Lactamase Inhibitor | 4.1 | 4.1 | -0.6% | 0.71 (0.70–0.72) |
| Cephalosporins (First Generation) | 7.7 | 7.6 | 1.5% | 0.68 (0.67–0.69) |
| Cephalosporins (Second/Third Generation) | 3.7 | 3.7 | -0.8% | 0.66 (0.64–0.67) |
| Fluoroquinolones (Second Generation) | 9.4 | 8.6 | 9.2% | 0.65 (0.64–0.66) |
| Fluoroquinolones (Third Generation) | 5.1 | 5.0 | 1.9% | 0.62 (0.61–0.63) |
| Macrolides | 9.3 | 9.3 | 0.5% | 0.73 (0.73–0.74) |
| Trimethoprim and/or Sulphonamides | 2.9 | 3.3 | -12.5% | 0.68 (0.67–0.69) |
| Nitrofurantoin | 2.7 | 2.8 | -3.3% | 0.69 (0.68–0.70) |
| Tetracyclines** | 1.4 | 0.6 | 116.4% | 0.46 (0.45–0.48) |
| Lincosamides | 1.0 | 1.0 | 5.4% | 0.74 (0.73–0.75) |
| Metronidazole | 1.5 | 1.5 | -0.5% | 0.68 (0.67–0.70) |
| Other | 0.3 | 0.3 | -20.0% | 0.90 (0.89–0.90) |

*(Xponent-ODB)/ODB x 100

**Doxycycline was not covered under the ODB program during this time period likely resulting in falsely low counts of tetracyclines within ODB; ODB = Ontario Drug Benefit

high antibiotic prescribing physicians were misclassified as being in the highest quartile of antibiotic prescribing volume, however only 6% of these physicians fell outside of the top 40% of prescribers. Furthermore, Xponent accurately captured the average number of antibiotic prescriptions per physician and the proportion of prolonged duration prescriptions, defined as >8 days. However, we identified an important limitation in the data with larger errors noted

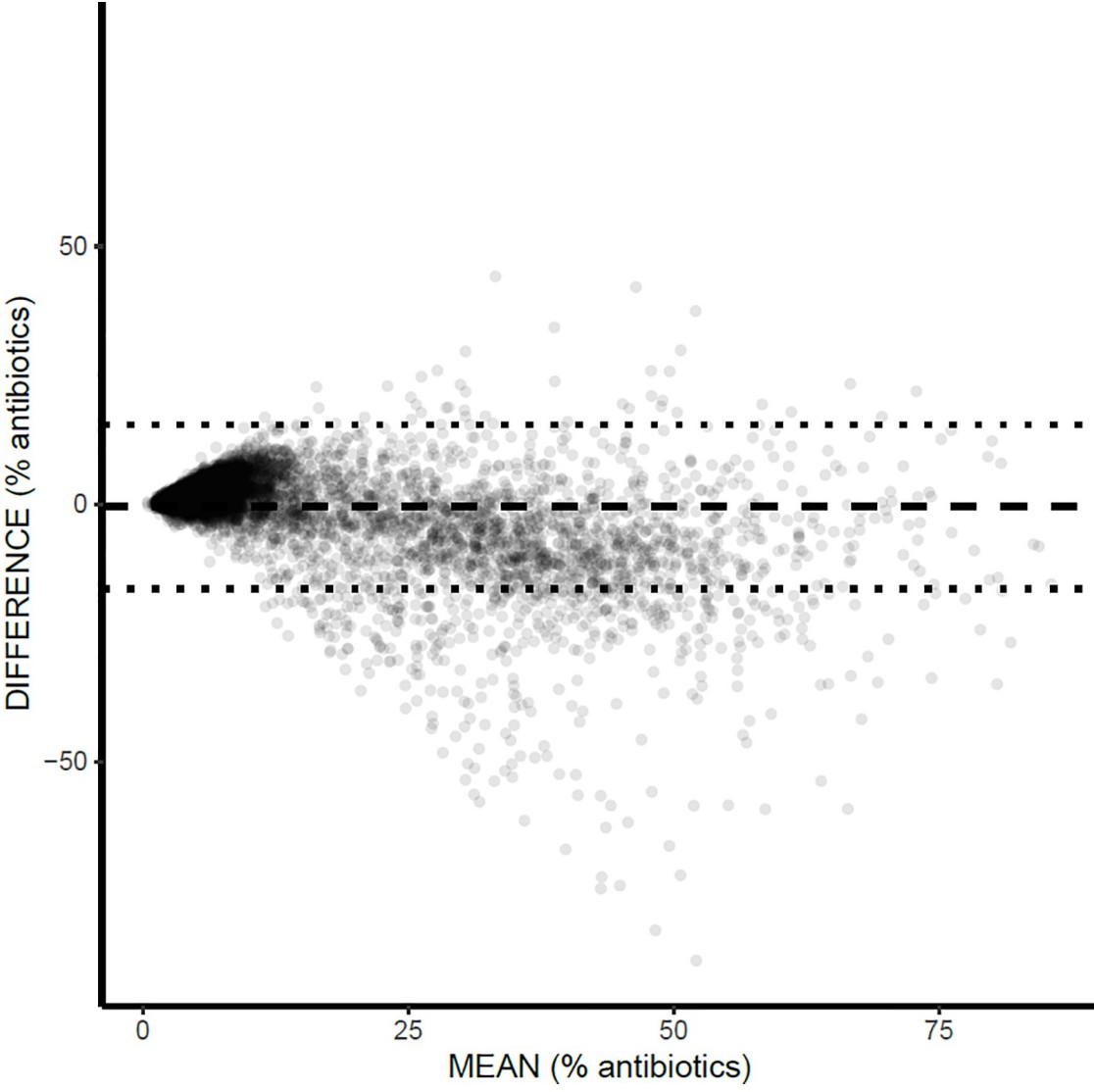

**Fig 3. Bland-Altman Plot from 9,272 physicians comparing the antibiotic rates (antibiotic prescriptions per 100 total medications prescribed) between Xponent and Ontario Drug Benefit (ODB) for female patients only.** Dash line = Mean difference (Xponent-ODB) = -0.5; Dotted lines = mean-2SD = -16.4 to mean+2SD = 15.4.

for physicians practicing in rural locations. This is most likely a reflection of a greater reliance on IQVIA's projection algorithm as they collect less prescription data directly from pharmacies in rural locations. This error was slightly more pronounced in female patients. Further exploration regarding differences in data validity between male and female patients may be warranted.

Tan *et al* demonstrated the validity of hospital antibiotic purchasing data from IQVIA as a reliable metric of antibiotic utilization compared to internal pharmacy records.[31] Abstracts validating IQVIA antibiotic databases in other Canadian jurisdictions, performed with patients of all ages, have been previously presented.[32,33] Lin *et al* demonstrated substantial variability between different databases in the U.S. for estimating antibiotic use, including a database from IMS (renamed as IQVIA), highlighting the importance of reliable data sources

**Table 4. Agreement between physician percent prolonged antibiotic prescription duration (defined as >8 days) quartiles in Xponent compared to the Ontario Drug Benefit (ODB) database.**

| Stratum | Mean percent prolonged duration per physician | | | Agreement between quartiles |
|---|---|---|---|---|
| | Xponent | ODB | Relative difference* | Weighted kappa (95% CI) |
| **Female Patients** | | | | |
| Total | 31.3 | 31.5 | -0.8% | 0.74 (0.73–0.75) |
| Rural region | 32.0 | 31.3 | 2.0% | 0.69 (0.66–0.72) |
| Urban region | 31.2 | 31.6 | -1.1% | 0.74 (0.73–0.75) |
| **Male Patients** | | | | |
| Total | 38.5 | 39.2 | -1.8% | 0.73 (0.73–0.74) |
| Rural region | 41.8 | 41.0 | 1.8% | 0.69 (0.66–0.72) |
| Urban region | 38.2 | 39.0 | -2.1% | 0.74 (0.73–0.75) |

*(Xponent-ODB)/ODB x 100; ODB = Ontario Drug Benefit

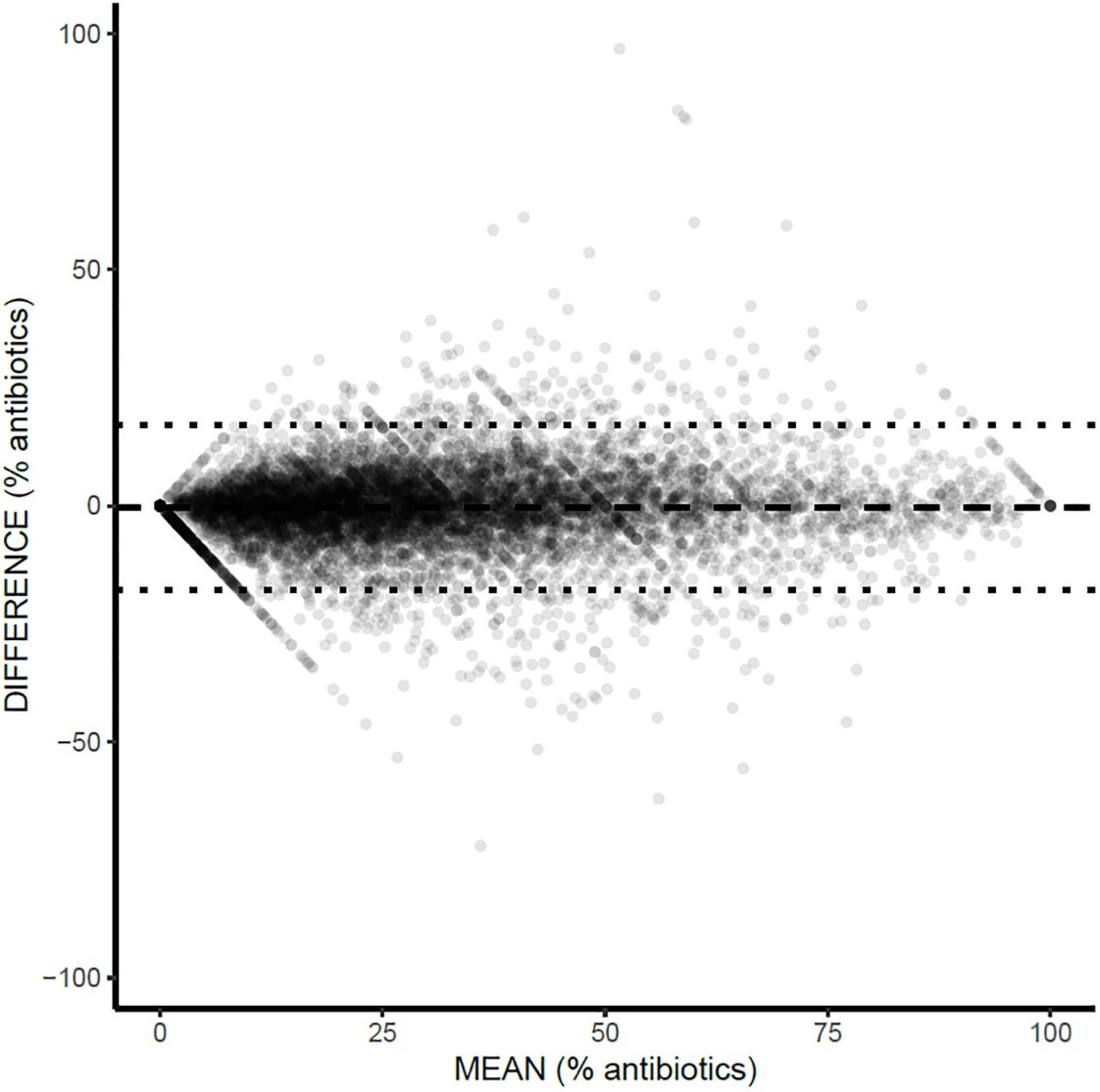

**Fig 4. Bland-Altman Plot from 9,272 physicians comparing the proportion of prolonged antibiotic duration (defined as >8 days) between Xponent and Ontario Drug Benefit (ODB) for female patients only.** Dash line = Mean difference (Xponent-ODB) = -0.3; Dotted lines = mean-2SD = -17.8 to mean+2SD = 17.1.

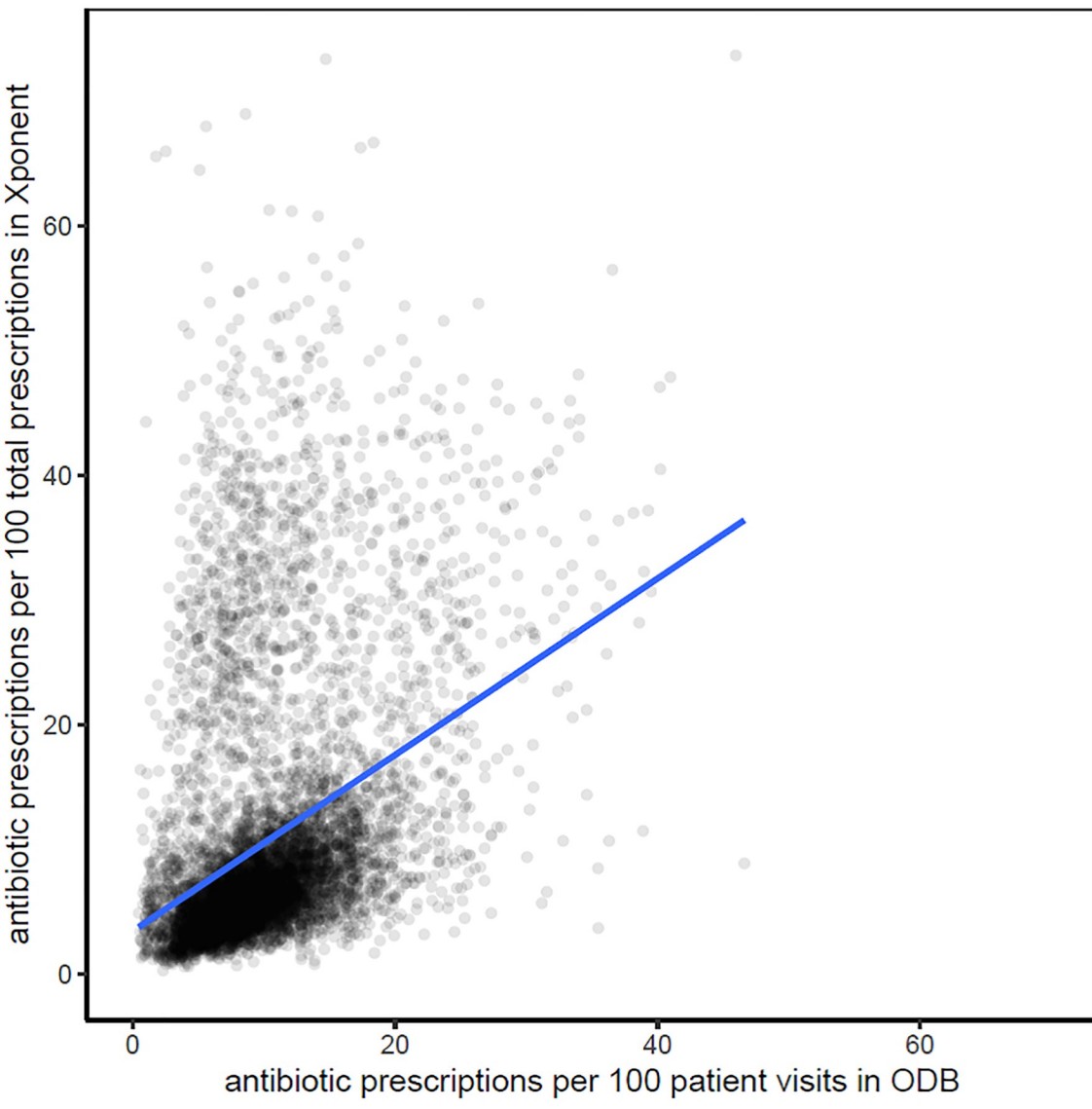

**Fig 5. Correlation of the number of antibiotic prescriptions per 100 total prescriptions in Xponent compared to antibiotic prescriptions per 100 patient visits in the Ontario Drug Benefit (ODB) database for female patients by primary care physicians.** Spearman correlation = 0.93; p<0.001.

for antibiotic use.[12] Both Canada and the U.S. rely on IQVIA for monitoring antibiotic use trends as well as assessing variability between regions and physicians.[7,13,15,24]

This study provides reassurance that IQVIA measures of total antibiotic prescribing in Xponent by physicians is valid. Furthermore, these results can assist public health departments, researchers, and policy makers towards appropriate uses of the data. We have demonstrated reasonable reliability of Xponent to identify high antibiotic prescribing physicians as well as reliably identifying physicians who prescribe prolonged antibiotic durations. Xponent is also quite reliable in estimating the number of antibiotic prescriptions per physician for both male and female patients. However, we caution the reliance on this database in rural areas or geographical locations where the data relies more heavily on the IQVIA projection algorithm. We demonstrated that antibiotic prescriptions per 100 total prescriptions is

imperfect, but is highly correlated with antibiotic prescriptions per 100 patient visits for primary care physicians in patients 65 years of age or older. This is a potentially useful metric from pharmacy data to indirectly account for differences in patient volume, when patient visits are unavailable.

This study has some limitations. While we demonstrated that the misclassification of prescribers in the top 25[th] percentile was small, this error may be important to clinicians receiving feedback on their prescribing. Certain subclasses of antibiotics were less well correlated between IQVIA and ODB databases, particularly trimethoprim and/or sulfamethoxazole and nitrofurantoin. One possible explanation for this could be the differences between the databases in defining repeats as these drugs are frequently used for prophylaxis for urinary tract infections in seniors. The IQVIA database utilized an explicit field from pharmacy data to denote refill prescriptions. No such field exists in ODB, and our study definition of repeats may have resulted in some differences in counting new antibiotic prescriptions. In addition, tetracyclines appeared to be overestimated in Xponent, however it is more likely that doxycycline was underestimated in ODB given that it was one of the only antibiotic treatments not funded by ODB during the study period. ODB does not fund First Nations populations with Non-insured Health Benefits, however this represents a small proportion of the included patient population. In Ontario, we only have complete population pharmacy data for patients 65 years of age or older. As a result validation studies may need to be conducted in other jurisdictions to determine whether our findings are generalizable to younger patient age groups. Non-physician prescribers are not included in either ODB or Xponent, and other databases are needed to study these important antibiotic prescribing populations.

In conclusion, we have validated the IQVIA Xponent antibiotic database to identify high prescribing physicians for patients 65 years of age and older, and identified some important limitations. Collecting accurate population-based antibiotic use data will remain vital to global efforts to combat rising rates of antimicrobial resistance. Governments and public health organizations should prioritize the need for accurate, population-based antimicrobial use datasets. An understanding of the uses and limitations of available databases are crucial for sound research and public policy decisions.

## Supporting information

**S1 Table. Oral outpatient antibiotic drugs included in the various antibiotic classes.**
(DOCX)

**S1 Fig. Bland-Altman plot from 9,272 physicians comparing the antibiotic rates (number of antibiotics prescribed per 100 total medications prescribed) between Xponent and Ontario Drug Benefit (ODB) for male patients only.** Dash line = Mean difference (Xponent-ODB) = -0.5; Dotted lines = mean-2SD = -16.0 to mean+2SD = 15.0.
(DOCX)

**S2 Fig. Bland-Altman plot from 9,272 physicians comparing the proportion of prolonged antibiotic duration (defined as >8 days) between Xponent and ODB for male patients only.** Dash line = Mean difference (Xponent-ODB) = -0.1; Dotted lines = mean-2SD = -21.0 to mean+2SD = 20.7.
(DOCX)

**S3 Fig. Correlation of antibiotic prescriptions per 100 total prescriptions in Xponent compared to antibiotic prescriptions per 100 patient visits in the Ontario Drug Benefit (ODB) database for male patients by primary care physicians.** Spearman correlation = 0.93;

p<0.001.
(DOCX)

## Acknowledgments

This study was supported by ICES, which is funded by an annual grant from the Ontario Ministry of Health and Long-Term Care (MOHLTC). The opinions, results and conclusions reported in this paper are those of the authors and are independent from the funding sources. No endorsement by ICES or the Ontario MOHLTC is intended or should be inferred.

## Author Contributions

**Conceptualization:** Kevin L. Schwartz, Bradley J. Langford, Kevin A. Brown, Nick Daneman, Jennie Johnstone, Julie HC Wu, Valerie Leung, Gary Garber.

**Data curation:** Kevin L. Schwartz.

**Formal analysis:** Cynthia Chen.

**Methodology:** Kevin L. Schwartz, Cynthia Chen, Bradley J. Langford, Kevin A. Brown, Nick Daneman, Jennie Johnstone, Valerie Leung, Gary Garber.

**Project administration:** Julie HC Wu.

**Supervision:** Gary Garber.

**Visualization:** Kevin A. Brown.

**Writing – original draft:** Kevin L. Schwartz.

**Writing – review & editing:** Cynthia Chen, Bradley J. Langford, Kevin A. Brown, Nick Daneman, Jennie Johnstone, Julie HC Wu, Valerie Leung, Gary Garber.

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
