## [Decision Letter · Decision Letter 0]

24 Jul 2019

PONE-D-19-17515

Validating a Popular Outpatient Antibiotic Database

PLOS ONE

Dear Dr. Schwartz,

Thank you for submitting your manuscript to PLOS ONE. After careful consideration, we feel that it has merit but does not fully meet PLOS ONE’s publication criteria as it currently stands. Therefore, we invite you to submit a revised version of the manuscript that addresses the points raised during the review process.

ACADEMIC EDITOR: 

Overall this manuscript addresses an important topic and has used a robust  methodology and chose to use Bland Altman plots which was an excellent choice. However there are certain issues to address to improve the manuscript.

Please respond to all the peer reviewer comments . In addition please respond to the following Academic Editor 's comments

1.Reviewer 2 noted 2 previous citations presented in poster format that did comparisons from 2 Canadian provinces which had comprehensive data ( all ages)  on outpatient prescriptions and one compared the data from the Xponent database so the authors need to rephrase their comments " to our knowledge community antibiotic use data has neve been externally validated.." which appears in a couple of settings in the manuscript.

2. You mention about a variety of prescription drug databases in the Introduction. Please add a couple of sentences to describe the commonly used ones and their differences eg Compuscript, the Canadian Disease and Therapeutic Index ,  National Prescription Audit  data and National Sales and Prescriptions data for the benefit of the readership.

3. Provide an estimate of NIHB prescriptions that may have been missed .

4. Please explicitly  explain if the prescriptions from non physician prescribers eg dentists, midwives, podiatrists etc were excluded reliably from the Xponent database which has obvious important implications

5. Please describe the geographic pharmacy difference if possible and what implications it may have

6. The Discussion is missing a limitations paragraph and there are several important ones - some mentioned earlier in the manuscript  but should be highlighted plus other mentioned about the limitations of the Xponenet database, non captured scripts eg NIHB, other prescribers, missing data and its influence and impact of a sensitivity analysis of > 14 days use as opposed to 8 .

7. There are several typos that need to be fixed eg refs 7-35 , presume it is 7, 35, missing words in some of the sentences

8. The references are very sloppy and full of errors ( too numerous to count )  in case, non use of urls and date of access, inappropriate case, spacing, non use of italics for Latin terms , incomplete references, missing references, inappropriate journal abbreviations and appear not to have been proof read by the authors very carefully . Please clean up ALL errors in the references.

We would appreciate receiving your revised manuscript by August 25th 2019. To enhance the reproducibility of your results, we recommend that if applicable you deposit your laboratory protocols in protocols.io, where a protocol can be assigned its own identifier (DOI) such that it can be cited independently in the future. For instructions see: http://journals.plos.org/plosone/s/submission-guidelines#loc-laboratory-protocols

We look forward to receiving your revised manuscript.

Kind regards,

John Conly, MD

Academic Editor

PLOS ONE

Journal Requirements:

3. Our internal editors have looked over your manuscript and determined that it is within the scope of our Antimicrobial Resistance call for papers. This collection of papers is headed by a team of Guest Editors for PLOS ONE: Kathryn Holt (Monash University and London School of Hygiene and Tropical Medicine), Alison H. Holmes (Imperial College London), Alessandro Cassini (WHO Infection Prevention and Control Global Unit), Jaap A. Wagenaar (Utrecht University). The Collection will encompass a diverse range of research articles; additional information can be found on our announcement page: https://collections.plos.org/s/antimicrobial-resistance. If you would like your manuscript to be considered for this collection, please let us know in your cover letter and we will ensure that your paper is treated as if you were responding to this call. If you would prefer to remove your manuscript from collection consideration, please specify this in the cover letter.

This study was funded by Public Health Ontario

The authors received no specific funding for this work

Reviewers' comments:

Reviewer's Responses to Questions

**Comments to the Author**

1. Is the manuscript technically sound, and do the data support the conclusions?

Reviewer #1: Yes

Reviewer #2: Yes

2. Has the statistical analysis been performed appropriately and rigorously? 

Reviewer #1: Yes

Reviewer #2: Yes

3. Have the authors made all data underlying the findings in their manuscript fully available?

Reviewer #1: Yes

Reviewer #2: No

4. Is the manuscript presented in an intelligible fashion and written in standard English?

Reviewer #1: Yes

Reviewer #2: Yes

5. Review Comments to the Author

Reviewer #1: Overall, this is a well written paper on a topic of considerable importance. Recognizing the need for reliable outpatient antibiotic utilization data, this study attempts to validate a pre-existing database that can be used to optimize outpatient antimicrobial stewardship. The study limitations were well recognized and stated clearly [ie. insufficient correlation between specific antibiotics and inability to validate for a) patients <65 years old, b) refill prescriptions]. The study also identified a weakness in the database with respect to the rural population which was well explained. The article's greatest strength is its rigorous statistical analyses.

A few additional comments for author response are:

1. With a NPV of 73%, 23% of prescribers were classified as belonging to the highest quartile of antibiotic prescribing when really they belonged to the top 40% according to OBD (the gold standard database). From an antimicrobial stewardship perspective, this discrepancy is acceptable because it still represents a physician group that should be targeted for further stewardship education and optimization of antibiotic use. However, from a clinician's perspective, being labeled as within the top 25% is very different from top 40%, the former signalling higher end of normal and the latter potentially being close to the mean. This NPV may not be acceptable to a clinician receiving their antibiotic prescribing summary.

2. Given a large subset of physicians are likely not represented (eg. rural physician etc), it would be interesting to see how many registered physicians within Ontario are excluded from the Xponent database entirely and therefore would not be subject to antibiotic utilization scrutiny. Is it possible to match the physicians to the College of Physicians and Surgeons of Ontario to obtain this number?

3. The authors performed multiple separate analyses for female vs. male genders and demonstrated that there were was a slightly greater relative difference in prescribing for male patients. Nevertheless, the discussions section lacked explanations on the potential reasons and significance of this finding. This should be further elaborated on.

4. The antibiotic prescribing rate "antibiotic prescriptions/100 total prescriptions" is an interesting metric. It is conceivable that a clinician whom is prescribing antibiotics excessively would likely prescribe other drugs excessively as well thereby obscuring the increased antibiotic use. It was reassuring to see that "antibiotic prescriptions/100 total prescriptions" correlated well with "antibiotic prescriptions / 100 patient visits" given the denominator for antibiotic utilization is typically patient visits to take into account differences in patient volume. It is not clear to me where the patient visit denominator is coming from (is it from OBD or an alternate source) and could these not just be incorporated into the numerator generated from Xponent rather than relying on the "antibiotic prescriptions/100 total prescriptions".

Reviewer #2: This topic is of great interest to likely a small group of individuals. IQVIA formerly Brogan, formerly IMS has recorded data on antimicrobial and other drug utilization from retail pharmacies for many years. The validity of the methods to extrapolate from the sample of pharmacies to the whole population has never been public and therefore the science community has been left to wonder how well their proprietary methods work and can they be trusted for research. In addition to the cited work of Tan (reference 42), two other groups in Canada have completed similar work in conference papers and could be cited (Dalton, B; Sabuda D, Bresee L et al. External Validation of Estimates of Antibacterial Dispensing in the IMS Brogan Xponent® Database in a Canadian Province. IDWeek 2013 https://idsa.confex.com/idsa/2013/webprogram/Paper41244.html) & Chong M, Dutil L, Bhatia T, Marra F, Patrick DM. Assessing antimicrobial consumption

using two different methodologies in British Columbia. Can J Infect Dis Med Microbiol

2007; 18(1): 35 Abstract A3.

In general, the authors have conducted a careful analysis using appropriate methods, and this manuscript is likely deserving of publication. I have a few suggestions to help make small improvements in clarity.

The goal of this study was to reliably predict the top 25 percentile of prescribers of antimicrobials to patients 65 and older, by identifying them in a public database of all pharmacies and comparing to the IQVIA database. I think the authors could do a better job describing this objective in the title as one does not reflexively think that quantification and validation would be performed at a physician level. I think some discussion of how these data would relate to the exponent database derived data on population level antimicrobial utilization would be appropriate (or if not at all, please state) and the conclusion "that the xponent database is validated for patients 65 and older " is actually inaccurate considering the principle finding of validating the identification of high prescibers of antibiotics in a population of 65 and older. "

The definition of physician antibiotic prescribing rate is unclear. Is this based on number of prescriptions, days of therapy or DDD etc.

Physicians were excluded for "prescribing less than 9 antibiotics". Does this mean <9 antibiotic prescriptions over the time period? Can you explain this number?

Could you define "antibiotics" in the study methods better? eg "systemic antibacterials" rather than just referring reader to the supplement.

The methods and results for agreement are confusing. In the methods it is stated the agreement of quartile groupings was evaluated, so one expects results in terms of categorical analysis. However in the text of results and table 3 mean number of of antibiotics prescriptions per physicians is reported.

The agreement on antibiotic prescribing per 100 prescriptions was assessed by Bland Altman plots. I am unclear of the relevance of antibiotic prescription per 100 prescriptions. This should be discussed in background and discussion.

If there are a significant number of non physician prescribers in Ontario, are they captured in ODB and exponent? Even if their prescribing rates are lower than that of physicians, it would be useful to note if agreement is similar with non physicians.

In figures 5 and S3 one can observe correlation but there appears to be bias with discounting of the xponent values. This is not commented upon. Can the slope of the regression line be provided? It would also provide easier interpretation if the x and y axes were given the same scales and number values.

6. PLOS authors have the option to publish the peer review history of their article (what does this mean?). If published, this will include your full peer review and any attached files.

Reviewer #1: No

Reviewer #2: Yes: Bruce Dalton

---

## [Author Response · Author response to Decision Letter 0]

8 Aug 2019

PONE-D-19-17515

Validating a Popular Outpatient Antibiotic Database

PLOS ONE

Dear Dr. Conly,

Thank you for the thoughtful review and helpful comments. We have incorporated all comments where possible and feel this has greatly improved the manuscript. We would like to be considered for the call for antimicrobial resistance papers. 

The data for this study is not publically available. The data set from this study is held securely in coded form at ICES. While data sharing agreements prohibit ICES from making the data set publicly available, access may be granted to those who meet pre-specified criteria for confidential access, available at www.ices.on.ca/DAS. The full data set creation plan and underlying analytic code are available from the authors upon request, understanding that the programs may rely upon coding templates or macros that are unique to ICES. The IQVIA Xponent dataset is owned and proprietary by IQVIA. The contract between PHO and IQVIA does not permit us to share the data publically. The authors had no special access privileges and other researchers may purchase the data from IQVIA directly (www.iqvia.com). 

Sincerely,

Kevin Schwartz

On behalf of the co-authors

 

ACADEMIC EDITOR: 

Overall this manuscript addresses an important topic and has used a robust methodology and chose to use Bland Altman plots which was an excellent choice. However there are certain issues to address to improve the manuscript.

Please respond to all the peer reviewer comments . In addition please respond to the following Academic Editor 's comments

1.Reviewer 2 noted 2 previous citations presented in poster format that did comparisons from 2 Canadian provinces which had comprehensive data ( all ages) on outpatient prescriptions and one compared the data from the Xponent database so the authors need to rephrase their comments " to our knowledge community antibiotic use data has neve been externally validated.." which appears in a couple of settings in the manuscript.

Response: Reference to this being the first validation study has been removed

2. You mention about a variety of prescription drug databases in the Introduction. Please add a couple of sentences to describe the commonly used ones and their differences eg Compuscript, the Canadian Disease and Therapeutic Index , National Prescription Audit data and National Sales and Prescriptions data for the benefit of the readership.

Response: The following sentences have been added to line 53: “Various forms of antibiotic use data are available including dispensing data compiled by pharmacies (e.g. National Prescription Audit® Compuscript® or GPM®)[7] or funders (e.g. Ontario Drug Benefit)[8] which can be population based. Other databases include dispensed data at the prescriber level (e.g. Xponent®) [9] or by prescription from electronic medical records.[10] These data sources have important differences and may not correlate over time. [11,12]”

3. Provide an estimate of NIHB prescriptions that may have been missed .

Response: We do not have precise numbers for this. There are approximately 10,000 registered Indian Status seniors living in Ontario according to StatsCan (https://www12.statcan.gc.ca/nhs-enm/2011/as-sa/99-011-x/99-011-x2011001-eng.cfm); roughly 2 million Ontario seniors are eligible for ODB. We feel this is unlikely to introduce significant bias at the physician level. We have added this line to the limitations line 257: “ODB does not fund First Nations populations with Non-insured Health Benefits, however this represents a small proportion of the included patient population.”

4. Please explicitly explain if the prescriptions from non physician prescribers eg dentists, midwives, podiatrists etc were excluded reliably from the Xponent database which has obvious important implications

Response: Non-physicians are not captured. We added this sentence line 93: “Non-physician prescribers (i.e. dentists, nurse practitioners, optometrists, etc.) are not included in this database.”

5. Please describe the geographic pharmacy difference if possible and what implications it may have

Response: We do not have granular data on the included and not included pharmacies from IQVIA. They will not share specific information but do admit that rural and smaller independent pharmacies are less likely to be included. This is reflected in the discussion line 221: “However, we identified an important limitation in the data with larger errors noted for physicians practicing in rural locations. This is most likely a reflection of a greater reliance on IQVIA’s projection algorithm as they collect less prescription data directly from pharmacies in rural locations.”

6. The Discussion is missing a limitations paragraph and there are several important ones - some mentioned earlier in the manuscript but should be highlighted plus other mentioned about the limitations of the Xponenet database, non captured scripts eg NIHB, other prescribers, missing data and its influence and impact of a sensitivity analysis of > 14 days use as opposed to 8 .

Response: The penultimate discussion paragraph has been edited to reflect the study's limitations (Line 247)

7. There are several typos that need to be fixed eg refs 7-35 , presume it is 7, 35, missing words in some of the sentences

Response: We have corrected all identified typos

8. The references are very sloppy and full of errors ( too numerous to count ) in case, non use of urls and date of access, inappropriate case, spacing, non use of italics for Latin terms , incomplete references, missing references, inappropriate journal abbreviations and appear not to have been proof read by the authors very carefully . Please clean up ALL errors in the references.

Response: Referencing formatting have been corrected to PLOS one standards and all errors have been corrected.

Journal Requirements:

Response: Manuscript modified to conform to these requirements

Response: Done

3. Our internal editors have looked over your manuscript and determined that it is within the scope of our Antimicrobial Resistance call for papers. This collection of papers is headed by a team of Guest Editors for PLOS ONE: Kathryn Holt (Monash University and London School of Hygiene and Tropical Medicine), Alison H. Holmes (Imperial College London), Alessandro Cassini (WHO Infection Prevention and Control Global Unit), Jaap A. Wagenaar (Utrecht University). The Collection will encompass a diverse range of research articles; additional information can be found on our announcement page: https://collections.plos.org/s/antimicrobial-resistance. If you would like your manuscript to be considered for this collection, please let us know in your cover letter and we will ensure that your paper is treated as if you were responding to this call. If you would prefer to remove your manuscript from collection consideration, please specify this in the cover letter.

Response: We would like to be considered for the collection, and have added to the cover letter above.

Response: We apologize for any confusion but ICES data and IQVIA data cannot be shared publically. We have provided the explanations in the cover letter above.

This study was funded by Public Health Ontario

The authors received no specific funding for this work

Response: Please change to: This study was funded by Public Health Ontario. We have removed this line from the manuscript 

Reviewers' comments:

Reviewer's Responses to Questions

Comments to the Author

1. Is the manuscript technically sound, and do the data support the conclusions?

Reviewer #1: Yes

Reviewer #2: Yes

2. Has the statistical analysis been performed appropriately and rigorously? 

Reviewer #1: Yes

Reviewer #2: Yes

3. Have the authors made all data underlying the findings in their manuscript fully available?

Reviewer #1: Yes

Reviewer #2: No

Response: Please see cover letter above for the explanation

4. Is the manuscript presented in an intelligible fashion and written in standard English?

Reviewer #1: Yes

Reviewer #2: Yes

5. Review Comments to the Author

Reviewer #1: Overall, this is a well written paper on a topic of considerable importance. Recognizing the need for reliable outpatient antibiotic utilization data, this study attempts to validate a pre-existing database that can be used to optimize outpatient antimicrobial stewardship. The study limitations were well recognized and stated clearly [ie. insufficient correlation between specific antibiotics and inability to validate for a) patients <65 years old, b) refill prescriptions]. The study also identified a weakness in the database with respect to the rural population which was well explained. The article's greatest strength is its rigorous statistical analyses.

A few additional comments for author response are:

1. With a NPV of 73%, 23% of prescribers were classified as belonging to the highest quartile of antibiotic prescribing when really they belonged to the top 40% according to OBD (the gold standard database). From an antimicrobial stewardship perspective, this discrepancy is acceptable because it still represents a physician group that should be targeted for further stewardship education and optimization of antibiotic use. However, from a clinician's perspective, being labeled as within the top 25% is very different from top 40%, the former signalling higher end of normal and the latter potentially being close to the mean. This NPV may not be acceptable to a clinician receiving their antibiotic prescribing summary.

Response: Thank for this comment. We have added this line to the limitations paragraph line 247): “While we demonstrated that the misclassification of prescribers in the top 25th percentile was small, this error may be important to clinicians receiving feedback on their prescribing.”

2. Given a large subset of physicians are likely not represented (eg. rural physician etc), it would be interesting to see how many registered physicians within Ontario are excluded from the Xponent database entirely and therefore would not be subject to antibiotic utilization scrutiny. Is it possible to match the physicians to the College of Physicians and Surgeons of Ontario to obtain this number?

Response: We cannot be certain if some physicians are not captured as we did not obtain all physicians in the province, only those that prescribed 1+ antibiotics in the time period. We did use CPSO numbers as the link between databases. Our linkage success rate was 97%. We clarified in line 168 that we used CPSO numbers: “The Xponent database had 25,678 unique physicians and 24,878 (97%) were successfully linked to ODB at ICES using College of Physicians and Surgeons identifiers.”

3. The authors performed multiple separate analyses for female vs. male genders and demonstrated that there were was a slightly greater relative difference in prescribing for male patients. Nevertheless, the discussions section lacked explanations on the potential reasons and significance of this finding. This should be further elaborated on.

Response: Thank you for this comment. For the most part we found the analysis for male and female patients to be similar. However there did appear to be a difference in validity between male and female patients in rural areas. Reasons for this are unclear and we have added this line to the discussion (line 224): “This error was slightly more pronounced in female patients. Further exploration for the differences between male and female patients may be warranted.”

4. The antibiotic prescribing rate "antibiotic prescriptions/100 total prescriptions" is an interesting metric. It is conceivable that a clinician whom is prescribing antibiotics excessively would likely prescribe other drugs excessively as well thereby obscuring the increased antibiotic use. It was reassuring to see that "antibiotic prescriptions/100 total prescriptions" correlated well with "antibiotic prescriptions / 100 patient visits" given the denominator for antibiotic utilization is typically patient visits to take into account differences in patient volume. It is not clear to me where the patient visit denominator is coming from (is it from OBD or an alternate source) and could these not just be incorporated into the numerator generated from Xponent rather than relying on the "antibiotic prescriptions/100 total prescriptions".

Response: Outpatient visits were obtained from the OHIP database. We have clarified this in the methods line 119. To do this comparison we linked IQVIA data to ICES. We feel it adds value to leave the prescriptions per 100 total prescription variable as some jurisdictions may not have the capacity to link to administrative data, and even for those that do it adds a layer of complexity. We completely agree with your analysis of the potential limitations of this variable and the implications for it from our study.

Reviewer #2: This topic is of great interest to likely a small group of individuals. IQVIA formerly Brogan, formerly IMS has recorded data on antimicrobial and other drug utilization from retail pharmacies for many years. The validity of the methods to extrapolate from the sample of pharmacies to the whole population has never been public and therefore the science community has been left to wonder how well their proprietary methods work and can they be trusted for research. In addition to the cited work of Tan (reference 42), two other groups in Canada have completed similar work in conference papers and could be cited (Dalton, B; Sabuda D, Bresee L et al. External Validation of Estimates of Antibacterial Dispensing in the IMS Brogan Xponent® Database in a Canadian Province. IDWeek 2013 https://idsa.confex.com/idsa/2013/webprogram/Paper41244.html) & Chong M, Dutil L, Bhatia T, Marra F, Patrick DM. Assessing antimicrobial consumption

using two different methodologies in British Columbia. Can J Infect Dis Med Microbiol

2007; 18(1): 35 Abstract A3.

Response: References added to background

In general, the authors have conducted a careful analysis using appropriate methods, and this manuscript is likely deserving of publication. I have a few suggestions to help make small improvements in clarity.

The goal of this study was to reliably predict the top 25 percentile of prescribers of antimicrobials to patients 65 and older, by identifying them in a public database of all pharmacies and comparing to the IQVIA database. I think the authors could do a better job describing this objective in the title as one does not reflexively think that quantification and validation would be performed at a physician level. I think some discussion of how these data would relate to the exponent database derived data on population level antimicrobial utilization would be appropriate (or if not at all, please state) and the conclusion "that the xponent database is validated for patients 65 and older " is actually inaccurate considering the principle finding of validating the identification of high prescibers of antibiotics in a population of 65 and older. "

Response: This point is well taken and correct. We have modified the title to: “Validating a Popular Outpatient Antibiotic Database to Reliably Identify High Prescribing Physicians”. We also modified the language in the abstract, objectives, and conclusions to more clearly reflect this (Lines 25, 43, 75, 264)

The definition of physician antibiotic prescribing rate is unclear. Is this based on number of prescriptions, days of therapy or DDD etc.

Response: We have clarified this in the methods line 99: “number of antibiotic prescriptions/number of total prescriptions by that physician”

Physicians were excluded for "prescribing less than 9 antibiotics". Does this mean <9 antibiotic prescriptions over the time period? Can you explain this number?

Response: Correct. We have clarified this in line 122 in the methods: “We excluded physicians who prescribed <9 antibiotics to patients ≥65 years of age during the study year in either Xponent or ODB databases. We used this cut off, which removes 5% of all antibiotic prescriptions, in order to exclude infrequent antibiotic prescribers.”

Could you define "antibiotics" in the study methods better? eg "systemic antibacterials" rather than just referring reader to the supplement.

Response: Modified to (line 96): “Antibiotics were defined as WHO ATC J01 class antibacterials for systemic use limited to those taken orally and dispensed from an outpatient pharmacy (S1 Table).”

The methods and results for agreement are confusing. In the methods it is stated the agreement of quartile groupings was evaluated, so one expects results in terms of categorical analysis. However in the text of results and table 3 mean number of of antibiotics prescriptions per physicians is reported.

Response: We have clarified in the methods what was done (line 142). We feel using the mean numbers helps with the clinical meaning of the kappas which are sometimes hard to interpret: “We descriptively compared the mean number of antibiotics prescribed per physician. Then we used the quartiles and evaluated agreement of the quartile groupings using weighted kappa statistics with 95% CIs for total antibiotics as well as subclasses of antibiotics.”

The agreement on antibiotic prescribing per 100 prescriptions was assessed by Bland Altman plots. I am unclear of the relevance of antibiotic prescription per 100 prescriptions. This should be discussed in background 

and discussion.

Response: We have clarified in the methods line 156: “IQVIA data does not contain numbers of patient visits. Therefore, we also assessed the correlation of the antibiotic prescribing rate (number of antibiotic prescriptions per 100 of total prescriptions) to number of antibiotic prescriptions per 100 patient visits using Spearman correlation coefficients.” We have also modified line 242 in the discussion to state: “We demonstrated that the number of antibiotic prescriptions per 100 total prescriptions is imperfect, but is highly correlated with the number of antibiotic prescriptions per 100 patient visits for primary care physicians in patients 65 years of age or older. This is a potentially useful metric from pharmacy data to indirectly account for differences in patient volume, when patient visits are unavailable.”

If there are a significant number of non physician prescribers in Ontario, are they captured in ODB and exponent? Even if their prescribing rates are lower than that of physicians, it would be useful to note if agreement is similar with non physicians.

Response: Non-physician prescribers are not captured in either database. We clarified this in the methods (lines 93 and 113) and added this to the limitations line 261. We do not have any data source in Ontario currently that captures non-physicians prescribers. 

In figures 5 and S3 one can observe correlation but there appears to be bias with discounting of the xponent values. This is not commented upon. Can the slope of the regression line be provided? It would also provide easier interpretation if the x and y axes were given the same scales and number values.

Response: Figures modified as suggested

---

## [Editor Report · Decision Letter 1]

27 Aug 2019

[EXSCINDED]

PONE-D-19-17515R1

Validating a Popular Outpatient Antibiotic Database to Reliably Identify High Prescribing Physicians

PLOS ONE

Dear Dr. Schwartz,

Thank you for submitting your manuscript to PLOS ONE and for addressing  the peer reviewer and editorial comments. However, there remain some additional issues which need to be addressed. Consequently, the manuscript does not fully meet PLOS ONE’s publication criteria as it currently stands. Therefore, we invite you to submit a revised version of the manuscript that addresses the points raised during the review process.

Thank you for the title change as per the reviewer comments - please add " for patients 65 years of age and older" as well to make this point even more explicit. Also mention this point in the sentence in the first paragraph of the Discussion.You have added the additional databases available in the Introduction as requested  but the references do not  reflect the actual databases - please apply more direct references and limit the your selection of self citations which are both  indirect and excessive.There are an excessive number of references cited in the Introduction about the point of both the CDC and Health  Canada  using the IQVIA data ie 17-39 and not all are necessary - please reduce the number by at least 50%.The sentence containing "  and the validity of IQVIA data has been assessed in some jurisdictions. [15] [16]" provides examples from within Canada and for broader populations than 65 and older and deserves to have  this point added. Alternatively it could be added to the  second paragraph of the Discussion when you speak about the use of this data in the US and Canada and add it as part of the academic discussion on the use of these databases.  There is a stray legend in italics on lines 204-205 just near the end of the Results - please correct this.Despite an explicit request, the Reference section continues to have multiple errors - incorrect format not in the standard style for date ( sometimes at the front of the reference in brackets and other times elsewhere and ), non use if urls where they should be present,  case errors everywhere and non use of italics for Latin terms etc.  - again too numerous to count. . Please correct  (do not rely on Mendeley which does not pick up all the formatting issues) or the manuscript will be returned again .

We would appreciate receiving your revised manuscript by Sept 16 2019 . To enhance the reproducibility of your results, we recommend that if applicable you deposit your laboratory protocols in protocols.io, where a protocol can be assigned its own identifier (DOI) such that it can be cited independently in the future. For instructions see: http://journals.plos.org/plosone/s/submission-guidelines#loc-laboratory-protocols

We look forward to receiving your revised manuscript.

Kind regards,

John Conly, MD

Academic Editor

PLOS ONE

---

## [Author Response · Author response to Decision Letter 1]

27 Aug 2019

August 27 2019,

Dear Dr. Conly,

Thank you for these comments. Each addressed below.

Sincerely,

Dr. Kevin Schwartz

On behalf of the co-authors

Comments:

• Thank you for the title change as per the reviewer comments - please add " for patients 65 years of age and older" as well to make this point even more explicit. Also mention this point in the sentence in the first paragraph of the Discussion.

Response: Added suggested line to title and line 215 in the discussion

• You have added the additional databases available in the Introduction as requested but the references do not reflect the actual databases - please apply more direct references and limit the your selection of self citations which are both indirect and excessive.

Response: The references were selected as examples for the aforementioned databases. Reference 7 used IQVIAs GPM database; We have added reference 8, which used CompuScript; reference 9 is an example of the ODB database; reference 10 used IQVIAs Xponent database; reference 11 used EMRALD (an administrative electronic medical record database). We have removed one of the EMRALD references.

• There are an excessive number of references cited in the Introduction about the point of both the CDC and Health Canada using the IQVIA data ie 17-39 and not all are necessary - please reduce the number by at least 50%.

Response: Reduced as requested

• The sentence containing " and the validity of IQVIA data has been assessed in some jurisdictions. [15] [16]" provides examples from within Canada and for broader populations than 65 and older and deserves to have this point added. Alternatively it could be added to the second paragraph of the Discussion when you speak about the use of this data in the US and Canada and add it as part of the academic discussion on the use of these databases. 

Response: Moved this point and references to Discussion paragraph 2, line 225 as suggested

• There is a stray legend in italics on lines 204-205 just near the end of the Results - please correct this.

Response: subheading removed as requested

• Despite an explicit request, the Reference section continues to have multiple errors - incorrect format not in the standard style for date ( sometimes at the front of the reference in brackets and other times elsewhere and ), non use if urls where they should be present, case errors everywhere and non use of italics for Latin terms etc. - again too numerous to count. . Please correct (do not rely on Mendeley which does not pick up all the formatting issues) or the manuscript will be returned again .

Response: References edited to specifications

---

## [Editor Report · Decision Letter 2]

6 Sep 2019

[EXSCINDED]

PONE-D-19-17515R2

Validating a Popular Outpatient Antibiotic Database to Reliably Identify High Prescribing Physicians for Patients 65 Years of Age and Older

PLOS ONE

Dear Dr. Schwartz,

Thank you for submitting your re-revised  manuscript to PLOS ONE and addressing most of the requested changes . There remain a couple of outstanding items that must be addressed.  Therefore, we invite you to submit a revised version of the manuscript that addresses the points raised during the review process.

ACADEMIC EDITOR: 

1. The sentence containing "and the validity of IQVIA data has been assessed in other  jurisdictions...…" should include the fact that  these assessments were done within Canada in patients of all ages....  which had been previously requested but  was not added  .Its placement in the Discussion is fine .

2. The references remain incomplete with several case and alignment errors. Please correct them as previously  requested.. 

We would appreciate receiving your revised manuscript by Oct 21 2019 11:59PM. To enhance the reproducibility of your results, we recommend that if applicable you deposit your laboratory protocols in protocols.io, where a protocol can be assigned its own identifier (DOI) such that it can be cited independently in the future. For instructions see: http://journals.plos.org/plosone/s/submission-guidelines#loc-laboratory-protocols

We look forward to receiving your revised manuscript.

Kind regards,

John Conly, MD

Academic Editor

PLOS ONE

---

## [Author Response · Author response to Decision Letter 2]

11 Sep 2019

1. The sentence containing "and the validity of IQVIA data has been assessed in other jurisdictions...…" should include the fact that these assessments were done within Canada in patients of all ages.... which had been previously requested but was not added .Its placement in the Discussion is fine .

 Response: Sentence modified as requested. To better reflect the references; it now reads: “Abstracts validating IQVIA antibiotic databases in other Canadian jurisdictions, performed with patients of all ages, have been previously presented.[32,33]” 

2. The references remain incomplete with several case and alignment errors. Please correct them as previously requested.. 

Response: Two people have independently reviewed the references for errors and they have been corrected

---

## [Editor Report · Decision Letter 3]

16 Sep 2019

Validating a Popular Outpatient Antibiotic Database to Reliably Identify High Prescribing Physicians for Patients 65 Years of Age and Older

PONE-D-19-17515R3

Dear Dr. Schwartz,

We are pleased to inform you that your manuscript has been judged scientifically suitable for publication and will be formally accepted for publication once it complies with all outstanding technical requirements.

With kind regards,

John Conly, MD

Academic Editor

PLOS ONE
---

## [Editor Report · Acceptance letter]

18 Sep 2019

PONE-D-19-17515R3 

Validating a Popular Outpatient Antibiotic Database to Reliably Identify High Prescribing Physicians for Patients 65 Years of Age and Older 

Dear Dr. Schwartz:

I am pleased to inform you that your manuscript has been deemed suitable for publication in PLOS ONE. Congratulations! Your manuscript is now with our production department. 

With kind regards,

on behalf of

Dr John Conly 

Academic Editor

PLOS ONE